# PPBoost: Progressive Prompt Boosting for Text-Driven Medical Image Segmentation

## Abstract

Text-prompted foundation models for medical image segmentation offer an intuitive way to delineate anatomical structures from natural language queries, but their predictions often lack spatial precision and degrade under domain shift. In contrast, visual-prompted models achieve strong segmentation performance across diverse modalities by leveraging spatial cues of precise bounding-box (bbox) prompts to guide the segmentation of target lesions. However, it is costly and challenging to obtain the precise visual prompts in clinical practice. We propose PPBoost (Progressive Prompt-Boosting), a framework that bridges these limitations by transforming weak text-derived signals into strong, spatially grounded visual prompts, operating under a strict zero-shot regime with no image- or pixel-level segmentation labels. PPBoost first uses a vision-language model to produce initial pseudo-bboxes conditioned on the textual object descriptions and applies an uncertainty-aware criterion to filter unreliable predictions. The retained image-bboxes pairs are then leveraged to train a pseudo-labeled detector, producing the high-quality bboxes for the query images. During inference, PPBoost further refines the generated bboxes by appropriately expanding them to tightly cover the target anatomical structures. The enhanced spatially-grounding bbox prompts guide existing segmentation models to generate final dense masks, effectively amplifying weak text cues into strong spatial guidance. Across three datasets spanning diverse modalities and anatomies, PPBoost consistently improves Dice and Normalized Surface Distance over text- and visual-prompted baselines and, notably, surpasses few-shot segmentation models without using labeled data. PPBoost can generalize to multiple typical visual segmentation model backbones. The anonymous code implementation is in: `https://anonymous.4open.science/r/submission-code-E2BB/`.

## 1 Introduction

Medical image segmentation assigns a semantic label to every pixel, yielding a dense mask that delineates anatomical structures and underpins diagnostic assessment Azad et al. (2024). Although supervised deep learning accompanied with image–mask pairs attains state-of-the-art accuracy, obtaining precise pixel-level annotations is costly and labor-intensive, especially in clinical settings that require expert knowledge Wang et al. (2022a). Weakly supervised segmentation Shen et al. (2023); Chen et al. (2022) has been explored to relax the need for dense mask annotations, which learns from coarse supervision such as image-level labels that can be collected at scale without pixel-wise tracing. Typically, these methods train an image-level classifier with global labels and apply class activation mapping (CAM) to derive class-specific saliency maps, which are then refined into pseudo masks for segmentation training. Despite their progress, this image-level supervision paradigm often captures only the most discriminative regions rather than the full anatomical structures. In addition, it is engineered for a single task or dataset, limiting their applicability across diverse medical imaging modalities.

Recent advances in large-scale foundation models (FMs) Kirillov et al. (2023); Lee et al. (2024); Zhao et al. (2025) offer a principled path to address these limitations by enabling promptable, generalizable, zero- or few-shot segmentation. Emerging FMs in medical image segmentation generally follow one of two prompting paradigms. *Visual-prompted models* rely on spatial cues (e.g., points, boxes) to generate the dense mask and have shown strong open-set generalization; no-

tably, MedSAM Ma et al. (2024) adapts Segment Anything Model (SAM) Kirillov et al. (2023) to medical imaging and deliver delivers powerful zero-shot segmentation performance with broad cross-scenarios robustness. *Text-prompted models* aim to segment directly from natural-language descriptions, providing a more intuitive and annotation-efficient interface. CLIP-based models such as BiomedCLIP Zhang et al. (2023b) and MedCLIP Wang et al. (2022b) have established vision–language priors and proven effective in biomedical retrieval, classification, and visual question answering. Extending this line, BiomedParse Zhao et al. (2025) enables text-conditioned segmentation by generating dense spatial confidence maps directly from prompts without relying on CAM and provides a natural bridge between text supervision and spatial delineation.

However, vanilla visual- or text-only prompting methods remains inadequate for the reliable medical image segmentation. First, visual-prompted models typically assume a precise spatial hint such as a bounding box (bbox) or a set of points tightly enclosing the target. While obtaining such prompts per case is costly, the small misplacements can cascade into large boundary errors. Second, the text-prompted approaches inherit CLIP's focus on global image–text alignment rather than dense localization. As illustrated in Fig. 2, such global alignment lead to weak spatial grounding, yielding coarse confidence maps (e.g., undersize, oversize, or even irrelevant segmentation results) for small or low-contrast lesions. These limitations are amplified under domain shift among different scanners, producing noisy masks and degraded performance in out-of-distribution (OOD) settings. These observations motivate us to propose following research question:

***How can we bridge the gap between spatially precise but costly visual prompts and intuitive yet weakly localized text prompts to achieve reliable, clinically intuitive medical image segmentation?***

To address these challenges, we introduce PPBoost, a Progressive Prompt-Boosting framework that converts weak text-derived signals into strong, spatially grounded visual prompts, enabling reliable medical image segmentation. Specifically, PPBoost converts textual object descriptions into robust bbox prompts through two successive stages. At training phase, as illustrated in Fig. 1, we exploit a vision–language model (VLM) to derive initial pseudo-bbox conditioned on the textual prompt of each training image. To improve reliability, we filter uncertain cases accompanied with high-variance pseudo-bbox predictions and then leverage the retained image-bbox data pairs to train an object detector, producing high-quality bbox-level supervision at scale. At inference stage, we adopt the trained detector to obtain pseudo-bboxes and propose bbox expansion procedure to further refine the spatial prompts, which are forwarded into a visual-prompted segmenter to generate the final dense masks. PPBoost yields reliable, use-intuitive segmentation from text-only supervision that can be readily provided by healthcare professionals. The main contributions are summarized below.

• We propose PPBoost, a weak-to-strong prompt transformation framework that converts coarse textual cues into precise, spatial visual prompts to guide medical image segmentation. PPBoost operates in a strict yet practical zero-shot regime: neither image-level nor pixel-level annotations are used to train the prompt generator or the segmentation model.

• We introduce a class of weak-to-strong prompt transformations that bridge textual cues to bbox-level prompts, including uncertainty-aware pseudo-bbox predictions with VLM, a detector trained on high-confidence image–bbox pairs for reliable supervision, and inference-time bbox refinement to ensure tight yet complete coverage of regions of interest.

• We validate our method on three challenging medical datasets spanning different imaging modalities and anatomical structures. Compared with the state-of-the-art of text- and visual-prompted segmentation counterparts, PPBoost consistently achieves superior performance, delivering average improvements of 6.69% and 7.32% in terms of mean Dice Similarity Coefficient and Normalized Surface Distance, respectively. Notably, PPBoost without reliance on labeled data outperforms the few-shot segmentation models. In addition, the visual prompts refined from PPBoost generalize well over typical medical segmentation models to accurately localize the target objects.

## 2 PPBoost: Text-Driven Medical Image Segmentation

We start by introducing a restricted target-aware text-driven medical image segmentation task designed to reflect real clinical scenarios. To address this task, we propose PPBoost, a weak-to-strong segmentation pipeline to convert the textual descriptions of target anatomical structures to visual bounding boxes and finally to dense segmentation masks.

## 2.1 Preliminaries of Segmentation and Foundation Models

**Task definition.** Given a medical image dataset with target anatomical structures (e.g.,tumor or organ) $\mathcal{I} = \{I_1, I_2, \ldots, I_N\}$, we consider a challenging but clinically relevant problem of zero-shot segmentation. In this setting, only a simple cue—such as a text description of an organ or a coarse spatial prompt—is provided to generate dense mask predictions. Among these cues, text queries are the most accessible in practice, but directly converting global text descriptions into dense masks remains extremely difficult and often results in weak spatial grounding.

**Text-prompted vision-language models.** Text-prompted VLMs aim to align medical images with natural language queries, enabling semantic concepts in text to be localized in image regions. A typical architecture consists of an *image encoder*, a *text encoder*, and a *joint embedding space* where the modalities are aligned via contrastive learning or cross-modal attention. Once trained, such models can localize regions of interest from simple text prompts. We build upon BiomedParse Zhao et al. (2025) to generate spatial confidence maps from text queries and retrieve the interested regions.

**Visual-prompted segmentation models.** An alternative paradigm is to guide segmentation with spatial prompts, such as points, bounding boxes, or regions of interest. Without loss of generalization, we leverage MedSAM Ma et al. (2024) as backbone model comprising an *image encoder* for dense features, a *prompt encoder* for spatial cues, and a *mask decoder* to fuse the two. The designed prompt boosting pipeline can adapt to other segmentation models as validated in Table 5.

**Motivation for PPBoost.** Taken together, text-prompted methods are easily accessible but weak in spatial grounding, whereas visual-prompted methods achieve strong segmentation but rely on costly annotations of precise bboxes. Their complementary strengths motivate our framework PPBoost, which consists of two successive stages: (i) a training pipeline that transforms textual inputs into pseudo-bboxes, and (ii) an inference pipeline that refines these bboxes informed by textual cues into high-quality spatial prompts for visual-prompted segmenters, yielding reliable dense masks.

## 2.2 Training: PPBoost for Text-Driven Pseudo-BBox Induction

As shown in Fig. 1, PPBoost first builds up a training pipeline to learn and strengthen correlations between text-informed medical images and pseudo-bboxes of target organs or lesions. This pipeline leverages a VLM to generate initial confidence maps used to extract pseudo-bboxes for each training image. Then, the uncertainty-aware filtering mechanism is proposed to aggregate confidence map predictions across model temperatures and filter out highly variant samples for fine-grained quality control. The retained samples are used to extract pseudo-bboxes of the target lesion, which serve as the upgraded labels to derive a lightweight detector, transforming global text prompts into bbox-level supervision. The implementation details are described below.

**Confidence map extraction for text-image consistency.** Considering a segmenting target (e.g., tumor, liver) of image dataset $\mathcal{I} = \{I_1, \ldots, I_N\}, I_i \in \mathbb{R}^{H \times W \times d}$, we use a high-performing commercial LLM (e.g., GPT–4 Achiam et al. (2023)) to generate $K$ sentence-level prompts $\mathcal{T} = \{T_1, \ldots, T_K\}$ describing images that contain the target segmented objects. We randomly assign prompts to images to form pairs $\mathcal{D} = \{(I_i, T_i)\}_{i=1}^N$. For each pair $(I_i, T_i)$, we evenly split $I_i$ into an $P_H \times P_W$ grid of non-overlapping patches and let $\Omega = \{1, \ldots, P_H \times P_W\}$ index these patches (so $j \in \Omega$ denotes a patch index). The VLM with encoders $\Phi_{\text{img}}$ and $\Phi_{\text{txt}}$ yields patch features $F_i = \Phi_{\text{img}}(I_i) \in \mathbb{R}^{P_H \times P_W \times d}$ and a text embedding $t_i = \Phi_{\text{txt}}(T_i) \in \mathbb{R}^d$. Let $f_{i,j} \in \mathbb{R}^d$ denote feature embedding of patch $j$, i.e., the $j$-th patch vector from $F_i$. We compute cosine-similarity logit between the text embedding and each patch feature as: $s_{i,j} = \langle f_{i,j}, t_i \rangle / (\|f_{i,j}\|_2 \|t_i\|_2), j = 1, \cdots, P_H \times P_W$. We then apply a spatial softmax over patches with temperature $\tau > 0$: $\tilde{S}_i(j) = \exp(s_{i,j}/\tau) / \sum_{j'} \exp(s_{i,j'}/\tau)$, where a smaller $\tau$ yields sharper maps. The normalized cosine-similarity logits are organized into a spatial confidence map $S_i \in \mathbb{R}^{P_H \times P_W}$ for subsequent bbox extraction, where each element indicates the possibility of a patch corresponding to the target segmentation lesion described in textual prompt.

**Uncertainty-aware confidence map filtering to select pseudo-bbox.** In practice, we observe that the frozen VLM yields confidence maps of varying quality: "easy" cases are sharp and well localized, whereas harder cases can be noisy or diffuse. Using all maps to extract pseudo-bboxes and train following bbox detector thus injects noise. We apply temperature scaling Lakshminarayanan et al. (2017); Minderer et al. (2021) to the cosine-similarity logits $s_{i,j}$ and form two spatial softmax

Figure 1: The pipeline of PPBoost. During training, we use VLM to obtain confidence maps and initial bboxes of text-prompted medical images and design filtering module to discard high-variance samples. The preserved images and bboxes are used to train a detector, producing reliable bboxes. During inference, we adopt the detector to regularize bboxes of text-prompted images and selectively expand them to obtain visual prompts, guiding segmentation model to predict masks.

maps over all patches $j' \in \Omega$:

$$\tilde{S}_{i,\text{low}}(j) = \frac{\exp\big(s_{i,j}/\tau_{\text{low}}\big)}{\sum_{j'\in\Omega}\exp\big(s_{i,j'}/\tau_{\text{low}}\big)}, \quad \tilde{S}_{i,\text{high}}(j) = \frac{\exp\big(s_{i,j}/\tau_{\text{high}}\big)}{\sum_{j'\in\Omega}\exp\big(s_{i,j'}/\tau_{\text{high}}\big)},$$

with $\tau_{\text{low}} < 1$ (sharper, high contrast) and $\tau_{\text{high}} \geq 1$ (smoother, lower contrast). A prediction is deemed reliable if it remains stable under this perturbation.

In practice, we retain cases with high agreement between $S_{i,\text{low}}$ and $S_{i,\text{high}}$ and discard the rest. Their discrepancy can be quantified by the KL divergence: $D_{\text{KL}}\big(S_{i,\text{low}}\|S_{i,\text{high}}\big) = \sum_{j\in\Omega} \tilde{S}_{i,\text{low}}(j) \log\big(\tilde{S}_{i,\text{low}}(j)/\tilde{S}_{i,\text{high}}(j)\big)$. Intuitively, a small value indicates stable, trustworthy predictions; a large value indicates instability. To implement hard filtering, we select a empirical threshold $\tau_{\text{KL}}$ and keep samples with $D_{\text{KL}} \leq \tau_{\text{KL}}$, filtering out the rest. For the retained samples, we use the smoother map $\tilde{S}_{i,\text{high}}$ to extract the pseudo-bboxs. Specifically, each $\tilde{S}_{i,\text{high}} \in \mathbb{R}^{P_H \times P_W}$ is firstly upsampled to the original image resolution ($H \times W$), then binarized into a foreground mask by applying a fixed threshold $\sigma$ (e.g. $\sigma=0.5$). From this binary mask, we extract the minimum enclosing rectangle that covers all activated foreground pixels as the pseudo-bbox $\bar{B}_i$. Consequently, a partially pseudo-labeled dataset is formed: $\bar{\mathcal{D}} = \{(\bar{I}_1, \bar{B}_1), \ldots, (\bar{I}_M, \bar{B}_M), \bar{I}_{M+1}, \ldots, \bar{I}_N\}$, where $\{(\bar{I}_i, \bar{B}_i)\}_{i=1}^{M}$ are the reliable image–box pairs and $\{\bar{I}_{M+1}, \ldots, \bar{I}_N\}$ are unlabeled images.

**Bbox detector training in a teacher-student framework.** Since the derived dataset $\bar{D}$ is only partially labeled, we adopt a classic semi-supervised learning paradigm, teacher–student framework Yang et al. (2022), to make full use of the supervision from both labeled and unlabeled samples, thereby deriving a stronger bbox detector. The bbox detector ingests the text-prompted images, regularizing the VLM-derived localizations and producing reliable bboxes for subsequent segmentation inference. In this framework, we first optimize the student parameters $\theta_s$ on the labeled subset under full supervision to initialize a stable teacher. During semi-supervised training, the student is optimized on (i) labeled data with a standard detection loss $L_{\text{sup}}$ (classification + bbox regression), and (ii) unlabeled data via consistency to teacher pseudo-labels. Concretely, the teacher predicts bboxes on *weak* augmentations and we retain only high-confidence predictions as pseudo-labels; the student then predicts on the corresponding *strong* augmentations and is trained to match these pseudo-labels, yielding $L_{\text{unsup}}$. The student is updated by backpropagation on $L_{\text{sup}} + \lambda L_{\text{unsup}}$, while the teacher is updated solely by exponential moving average (EMA) Haynes et al. (2012): at iteration $k$, the parameters of the teacher model are calculated as $\theta_t^{(k)} = \alpha\,\theta_t^{(k-1)} + (1-\alpha)\,\theta_s^{(k)}$ with decay factor $\alpha \in (0,1)$.

## 2.3 INFERENCE: PPBOOST FOR BBOX REFINEMENT AND MASK GENERATION

After training the bbox detector, PPBoost uses it to infer the pseudo-bboxes for text-prompted medical images (Fig. 1). To further enhance prompt quality, we apply a simple yet effective refinement step that selectively expands the predicted boxes to ensure tight yet complete coverage of the target region. The refined boxes are then used as visual prompts for a segmentation model, which produces the final dense masks.

**Selective expansion of bboxes.** Since the detector is trained on pseudo-bboxes generated by a text-driven model rather than ground-truth annotations, it inevitably learns from noisy supervision and, as a result, produces noisy bboxes at inference. According to their relative position with respect to the ground-truth boxes, we categorize these pseudo-bboxes into four types, as illustrated in Fig. 2: (1) *high-quality* boxes that tightly cover most of the target region; (2) *undersized* boxes that miss part of the target and suffer from discriminative region issues; (3) *oversized* boxes that include the target with a large surrounding margin; and (4) *irrelevant* boxes that are far from the target.

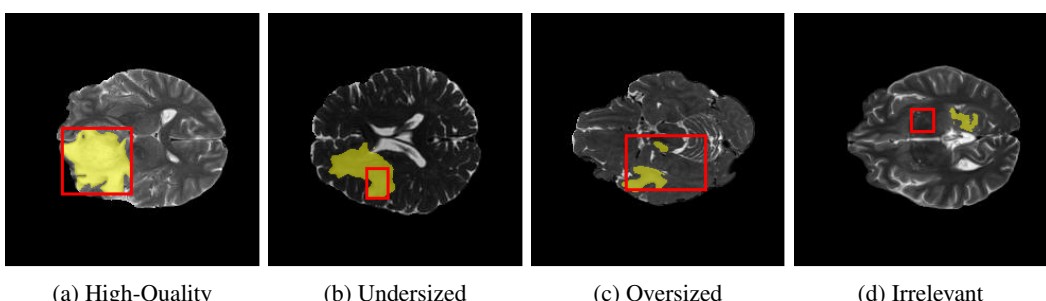

| (a) High-Quality | (b) Undersized | (c) Oversized | (d) Irrelevant |

Figure 2: Visualizations of different types of pseudo-bboxes on the BraTS 2021 dataset.

Empirically, undersized boxes dominate the detector's outputs owing to our uncertainty-aware filtering and selective retention of high-confidence pseudo-bboxes that favors precision over recall. This observation suggests a simple refinement strategy: expanding undersized pseudo-bboxes so they cover more of the target region, while leaving other bboxes unchanged. The challenge lies in the fact that we cannot directly identify whether a predicted bbox is undersized or other types without comparing against ground-truth labels. This motivates the following question: "If we expand both undersized bboxes and others by a small factor, would the benefit from correcting undersized boxes outweigh the potential degradation from perturbing high-quality ones?" To investigate this, we conduct controlled experiments where the undersized boxes are simulated by shrinking the selected high-quality bboxes by 10%, 15%, and 20%. Similarly, we simulate the oversized boxes by expanding them by the same ratios in independent trials. Results, shown in the Appendix A, reveal that expanding the undersized boxes significantly improves segmentation accuracy, while expanding other types of bboxes causes little to no degradation in visual-prompted segmentation models. This is because the expansion size is modest and visual-prompted segmenters tolerate slight over-coverage, expanding non-undersized boxes has negligible adverse effect.

Based on these findings, we design a selective expansion strategy. For a test image $I_i$ with detector output $\hat{B}_i = (x_i, y_i, w_i, h_i, s_i)$, where $(x_i, y_i)$ denote the top-left corner, $(w_i, h_i)$ the width and height, and $s_i$ the confidence score, we selectively refine the predicted bboxes. While bboxes with higher detection confidence scores ($s_i > \varphi$) are more likely to be high-quality and are therefore kept unchanged, boxes with lower confidence scores are expanded outward by a fixed ratio $r$. Formally:

$$(x_i', y_i', w_i', h_i') = \begin{cases} (x_i, y_i, w_i, h_i), & s_i > \varphi, \\ (x_i - \frac{r}{2}w_i, \ y_i - \frac{r}{2}h_i, \ (1+r)w_i, \ (1+r)h_i), & s_i \le \varphi. \end{cases}$$

**Visually prompted image-to-mask.** Building on the expanded prompt, we adopt a general visually prompted segmentation model with an image encoder $E_{\mathrm{img}}$, a prompt encoder $E_{\mathrm{prm}}$, and a mask decoder $D_{\mathrm{mask}}$. Given $I_i$ and $\hat{B}_i'$, we compute image features $z_i = E_{\mathrm{img}}(I_i)$ and a prompt embedding $p_i = E_{\mathrm{prm}}(\hat{B}_i')$. The decoder outputs mask logits $\ell_i = D_{\mathrm{mask}}(z_i, p_i) \in \mathbb{R}^{H \times W}$. Conditioned on the mask logits, we obtain the probability map $\hat{P}_i = \sigma(\ell_i) \in [0, 1]^{H \times W}$ and then binarize at a standard segmentation threshold $\tau_{\mathrm{seg}} = 0.5$ to produce the final mask prediction $M_i = \mathbf{1}\{\sigma(\ell_i) > \tau_{\mathrm{seg}}\}$, thereby completing the image-to-mask generation pipeline.

## 3 EXPERIMENTS

**Datasets.** We conduct a comprehensive evaluation of the proposed PPBoost pipeline on three benchmark medical image datasets, including the BraTS 2021 Baid et al. (2021), LiTS 2017 Bilic et al.

Table 1: Performance comparison on medical image segmentation. Our method, PPBoost, is compared against baselines across several text-driven and visual-driven methods. We report mDSC and mNSD in percentages (%). The best performance in each column is highlighted in **bold**.

| Setting | Methods | BraTS21 | | LiTS17 | | KidneySeg | | Average | |
|---------|---------|---------|---------|---------|---------|---------|---------|---------|---------|
| | | mDSC | mNSD | mDSC | mNSD | mDSC | mNSD | mDSC | mNSD |
| *Visual-driven* | UniverSeg (5 shots) | 30.62 | 37.67 | 60.66 | 63.90 | 78.93 | 83.40 | 56.74 | 61.66 |
| | FFSP-SAM (5 shots) | 16.04 | 19.20 | 53.60 | 55.41 | 59.99 | 64.75 | 43.21 | 46.45 |
| | UniverSeg (10 shots) | 45.22 | 54.03 | 63.84 | 66.87 | 85.65 | 89.59 | 64.90 | 70.16 |
| | FFSP-SAM (10 shots) | 15.30 | 18.43 | 57.67 | 59.65 | 64.79 | 70.04 | 45.92 | 49.37 |
| | SAMAug | 18.48 | 21.42 | 24.26 | 25.15 | 31.88 | 33.56 | 24.87 | 26.71 |
| *Text-driven* | SaLiP | 20.43 | 23.85 | 39.82 | 41.15 | 13.81 | 15.13 | 24.69 | 26.71 |
| | nnU-Net | 59.60 | 68.52 | 60.67 | 62.58 | 84.60 | 87.89 | 68.29 | 72.30 |
| | MedCLIPSAMv2 | 35.90 | 41.13 | 15.92 | 17.16 | 6.15 | 7.41 | 19.32 | 21.9 |
| *Text-to-visual* | **PPBoost** | **60.71** | **69.31** | **74.10** | **76.32** | **90.14** | **93.24** | **74.98** | **79.62** |

(2023), and CT2USforKidneySeg Song et al. (2022). For brevity, we refer to them as BraTS21, LiTS17, and KidneySeg in the following. The dataset details are listed in Appendix A.1.

**Implementation details.** 1) *training stage:* We employ a recently released radiology-specialized model, Biomedparse, as the text-prompted model backbone to generate pseudo-bboxes in our pipeline. We then use GPT-4 to generate a prompt pool containing 20 text phrases about the target anatomical structure for each dataset, where each image is paired with a randomly selected text prompt. In the confidence map filtering, two temperatures are adopted: $\tau_{low} = 0.1$, and $\tau_{high} = 1$. We set the filtering threshold to retain the bottom 30% of samples ranked by KL-divergence and use a standard threshold 0.5 to convert those reliable confidence maps into pseudo-bbox labels. In the detector training, we adopt the unbiased teacher framework based on the Faster R-CNN backbone and randomly select 10% of the extracted pseudo-bboxes as labels to avoid overfitting. The detector training is performed with size of 32 labeled and 32 unlabeled images per batch, using a base learning rate of 0.004 and a maximum of 10000 iterations. The EMA decay rate for teacher updates is set to 0.9996. We use a burn-in phase of 800 iterations before unsupervised learning begins, and weight the unsupervised consistency loss by 1.5. Non-maximum suppression (NMS) is applied with an IoU threshold of 0.7 to remove duplicate detections. The mean Average Precision (mAP), $mAP_{50}$ and $mAP_{75}$ are used for evaluating the detection performance. 2) *inference stage:* We set the bbox score threshold in selective expansion to the median confidence across the inference set. For the downstream segmentation stage, we use the MedSAM as the visual-prompted segmentor in our pipeline. The mean Dice Similarity Coefficient (mDSC) and mean Normalized Surface Dice (mNSD) are used as metrics to provide a comprehensive comparison with other baseline methods. All experiments are conducted on NVIDIA L40s GPUs.

### 3.1 COMPARISON WITH THE BASELINES

We compare our proposed PPBoost with its segmentation counterparts in different settings, including visual-driven and text-driven segmentation methods. The visual-driven counterparts include ground-truth point prompt method of SAMAug Dai et al. (2023), few-shot ground-truth mask prompt methods of UniverSeg Butoi et al. (2023), and Few-shot Self-Prompt SAM (FFSP-SAM) Wu et al. (2023). For text-driven counterparts, we present the CLIP-based text to bbox prompt generation methods: SaLIP Aleem et al. (2024) and MedCLIP-SAMv2 Koleilat et al. (2025). In addition, we construct a nnU-Net Isensee et al. (2021) baseline by training the nnU-Net segmenter in a fully supervised manner on the raw mask outputs produced by the BiomedParse.

**Observation (Obs.) 1: As shown in Table 1, our proposed PPBoost consistently outperforms both visual-driven and text-driven methods.** Compared to the best visual-driven baseline, UniverSeg trained with 10 shots of ground-truth masks, PPBoost achieves average gains of +10.08% mDSC and +9.46% mNSD across the three datasets, while requiring no spatial annotations or inputs. For text-driven methods, the previous state-of-the-art MedCLIP-SAMv2 generalizes poorly on all the datasets, whereas PPBoost demonstrates robust performance. Notably, nnU-Net segmenter

Figure 3: Visualization of the segmentation results on BraTS21, LiTS17 and KidneySeg datasets.

is trained directly on pseudo masks. In contrast, PPBoost learns solely from filtered pseudo-bboxes to train the detector and uses the visual prompt to inform fixed MedSAM generating masks at inference. Despite this weaker supervision, PPBoost surpasses nnU-Net on all the datasets—by +1.11% mDSC / +0.79% mNSD on BraTS21, +13.43% / +13.74% on LiTS17, and +5.54% / +5.35% on KidneySeg—highlighting the effectiveness of our progressive prompt design in converting noisy textual cues into reliable visual bboxes while reducing training cost. Fig. 3 shows visual examples of segmentation results produced by different methods on the three datasets.

We further investigate the relationship between detection quality and segmentation performance of PPBoost. The bbox detection performance is listed in Table 2, reporting mAP, $AP_{50}$, and $AP_{75}$ scores. In particular, KidneySeg achieves the highest $AP_{50}$ and thus the strongest relative segmentation performance, whereas BraTS21 exhibits the lowest $AP_{50}$ and the weakest relative segmentation. **Obs.2: These results demonstrate that segmentation performance in PPBoost is tightly coupled with detection quality.** Specifically, $AP_{50}$ serves as the most reliable indicator of downstream performance compared to stricter metrics such as mAP and $AP_{75}$. This is because reliable coarse localization quality is more critical for MedSAM prompting than fine-grained bounding box tightness.

Table 2: Bbox detection accuracy performance of PPBoost in percentage.

| Dataset | mAP | $AP_{50}$ | $AP_{75}$ |
|---------|-----|-----------|-----------|
| BraTS21 | 32.84 | 66.30 | 28.56 |
| LiTS17 | 55.63 | 75.42 | 64.86 |
| KidneySeg | 52.70 | 91.49 | 50.89 |

### 3.2 GENERALIZATION AND ROBUSTNESS OVER DIFFERENT CONFIGURATIONS

To evaluate the proposed pipeline's generalization and robustness capabilities, we test PPBoost with diverse detector backbones, visual-prompted foundation models, and model hyperparameters.

**Semi-supervised detector backbones.** We employ Unbiased Teacher as the default bbox detector, as it provides a strong and stable Semi Supervised Object Detector (SSOD) baseline while maintaining training efficiency. To evaluate the robustness of our framework with respect to this choice, we incorporate two alternative

Table 3: Segmentation performance (mDSC, %) of different SSOD backbones within the PPBoost pipeline.

| Backbone | BraTS21 | LiTS17 | KidneySeg | Avg. |
|----------|---------|--------|-----------|------|
| Unbiased Teacher | 60.71 | 74.10 | 90.14 | 74.98 |
| Semi-DETR | 58.45 | 72.89 | 90.05 | 73.80 |
| Soft Teacher | **61.13** | **78.73** | **90.63** | **76.83** |

SSOD methods: Soft Teacher Xu et al. (2021) and Semi-DETR Zhang et al. (2023a). Table 4 reports the detection accuracy of the three backbones. From the detection perspective, their detection performance varies under different metrics. Semi-DETR achieves the highest overall mAP on LiTS17 and KidneySeg, while Soft Teacher produces the strongest $AP_{50}$ values on all the datasets. From the segmentation perspective, as shown in Table 3, Soft Teacher delivers the best Dice performance across all datasets. This aligns with our earlier observation that segmentation quality

Table 4: Comparison of different SSOD backbones within the PPBoost pipeline across three datasets. We report detection performance in terms of mAP, $AP_{50}$, and $AP_{75}$ (%).

| Backbone | BraTS21 | | | LiTS17 | | | KidneySeg | | |
|---|---|---|---|---|---|---|---|---|---|
| | mAP | $AP_{50}$ | $AP_{75}$ | mAP | $AP_{50}$ | $AP_{75}$ | mAP | $AP_{50}$ | $AP_{75}$ |
| Unbiased Teacher | 32.8 | **66.3** | 28.6 | 55.6 | 75.4 | 64.9 | 52.7 | 91.5 | 50.9 |
| Soft Teacher | 36.2 | **66.3** | 35.6 | 58.5 | **79.2** | 66.5 | 58.0 | **94.6** | 67.1 |
| Semi-DETR | 31.6 | 59.0 | 29.9 | 63.6 | 76.0 | 67.7 | 58.7 | 89.5 | 60.9 |

Table 5: Segmentation result (mDSC%) across various visual prompt models. "Direct" refers to using the pseudo-bboxes to directly prompt segmentation models without detector training.

| Segmentor | BraTS21 | | LiTS17 | | KidneySeg | |
|---|---|---|---|---|---|---|
| | Direct | PPBoost | Direct | PPBoost | Direct | PPBoost |
| SAM | 50.92 | **57.15** | 58.70 | **73.25** | 74.24 | **81.88** |
| SAM-Med2D | 39.22 | **44.76** | 49.22 | **61.09** | 68.47 | **73.92** |
| MedSAM | 48.05 | **60.71** | 60.78 | **74.10** | 82.93 | **90.14** |

correlates most consistently with $AP_{50}$. **Obs.3: Overall, these results demonstrate that PPBoost is detector-agnostic, as it maintains strong performance across different SSOD backbones.**

**Visual-prompted foundation models.** While MedSAM is adopted as the default visual-prompted segmentation models, we also replace it with SAM Kirillov et al. (2023) and SAM-Med2D Cheng et al. (2023b). For each model, we evaluate the final segmentation results in two different settings: (i) directly using the pseudo-bboxes produced by BiomedParse as prompts, and (ii) using the refined bbox prompts generated by the PPBoost pipeline. **Obs.4: As illustrated in Table 5, across all three visual-prompted models, the PPBoost-generated bbox prompts consistently yield significantly better segmentation results than directly using raw pseudo-bboxes.** This phenomenon indicates that our framework does not rely on a specific visual-prompted segmentation model. Instead, the refined bbox prompts produced by PPBoost provide a general advantage, serving as a stronger and more reliable input to different foundation models.

**KL divergence threshold in confidence map filtering.** We further examine the robustness of PPBoost with respect to the KL-divergence filtering threshold. The pseudo-bboxes used to train detector are extracted from a filtered subset of confidence maps, which can be obtained retaining samples at different thresholds. We use the bottom 15%, 30% (default), or 50% of samples ranked by KL-divergence to test robustness of PPBoost. Table 6 reports the segmentation performance across BraTS, KidneySeg, and LiTS datasets.

**Obs.5: The results reveal that a moderate threshold is crucial.** Overly strict filtering (e.g., 15%) yields higher-quality pseudo-labels but sacrifices diversity, which hurts generalization. Conversely, looser filtering (e.g., 50%) retains more diverse samples but introduces noise, leading to degraded performance in some cases. Notably, sensitivity differs across datasets: LiTS achieves its best performance at 50%, indicating

Table 6: Segmentation results (mDSC, %) under different KL-divergence filtering thresholds.

| Threshold | BraTS | Kidney | LiTS | Avg. |
|---|---|---|---|---|
| 50% | 59.14 | 87.32 | **76.68** | 74.38 |
| 15% | 58.88 | 89.00 | 67.72 | 71.87 |
| 30% (default) | **60.71** | **90.14** | 74.10 | **74.98** |

that diversity outweighs strict quality, while BraTS and KidneySeg perform best at 30%, suggesting that balancing sample diversity and label quality is most beneficial.

**Ablation study.** We conduct detailed ablation experiments on key components of the proposed PPBoost and present the results in Table 7, which leads to **Observation 6**: First, adding the detector (i.e., the PPBoost training stage) markedly improves segmentation, yielding an average gain of $+11.06\%$ mDSC over directly prompting with the pseudo-bboxes derived from Biomedparse. Second, removing the self-ensemble filtering module reduces performance by an average of $3.85\%$ mDSC (up to $8.03\%$ on BraTS21), underscoring its role in mitigating pseudo-label noise. Finally, selective expansion provides an additional $+0.89\%$ average mDSC improvement, consistent with Sec. 2.3: enlarging high-quality boxes has little effect, whereas expanding undersized boxes offers substantial benefits.

Table 7: Left columns indicate included modules: Teacher-student Detector, Self-ensemble Filtering, Selective Expansion. A checkmark ✓ indicates that the corresponding module is enabled, whereas a blank cell denotes that the module is ablated. Numbers are mDSC (%); Avg. is the mean over datasets. Best in **bold**, second-best underlined.

| Detector | Filtering | Expansion | BraTS21 | LiTS17 | KidneySeg | Avg. |
|---|---|---|---|---|---|---|
| | | | 48.05 | 60.78 | 82.93 | 63.92 |
| ✓ | | ✓ | 52.68 | 72.06 | 88.65 | 71.13 |
| ✓ | ✓ | | 58.78 | 73.62 | 89.88 | 74.09 |
| ✓ | ✓ | ✓ | **60.71** | **74.10** | **90.14** | **74.98** |

## 4 RELATED WORK

**Visual-Prompted Foundation Model in Medical Image Domain.** As a generic visual prompt-based foundation model, Segment Anything Model (SAM) Kirillov et al. (2023) is designed with a robust image encoder, a prompt encoder, and a lightweight mask decoder. Trained on over 1 billion masks from 11 million natural images, SAM shows strong zero-shot segmentation when given point or bounding box prompts. Building on this, numerous methods Cheng et al. (2023a); Ma et al. (2024); Shaharabany et al. (2023); Huang et al. (2023) have been developed to adapt SAM for universal medical image segmentation. MedSAM Ma et al. (2024) fine-tunes only the lightweight mask decoder with a large-scale medical data and bounding box inputs. while SAM-Med2D Cheng et al. (2023a) introduces learnable adapters to the image encoder for domain-specific transfer without discarding pretrained features.

**Text-Prompted Foundation Model in Medical Image Domain.** Text-prompted foundation models have gained increasing attention in medical image analysis due to their ability to incorporate clinical language. Given CLIP Radford et al. (2021)'s strong generalization and zero-shot adaptation in text-image alignment, several medical variants have been developed. PubMedCLIP Eslami et al. (2023), fine-tuned on PubMed image–text pairs; MedCLIP Wang et al. (2022b), which decouples image–text training to handle unpaired data; and BiomedCLIP Zhang et al. (2023b), pre-trained on large-scale PubMed Central corpora and achieving state-of-the-art classification and retrieval performance. However, these CLIP-based adaptations show limited generalization in object localization. To address this, BiomedParse Zhao et al. (2025) was recently introduced as a foundation model capable of directly grounding diverse medical images from text queries.

**Semi-supervised Object Detection.** Semi-supervised object detection reduces annotation costs by leveraging a small labeled set with a large pool of unlabeled data. A dominant paradigm is the teacher–student framework Tang et al. (2021); Liu et al. (2021); Xu et al. (2021); Li et al. (2022); Zhang et al. (2023a); Xu et al. (2023), where the student is trained on labeled data and a teacher, updated via Exponential Moving Average (EMA), generates pseudo-labels for unlabeled samples under consistency constraints. Performance depends critically on pseudo-label quality and training robustness. Representative advances include loss reweighting for noisy or imbalanced labels (Unbiased Teacher Liu et al. (2021)), localization refinement strategies (Soft Teacher Xu et al. (2021), PseCo Li et al. (2022)), and architectural extensions such as Semi-DETR Zhang et al. (2023a), which adapts transformer-based detectors to the semi-supervised setting and achieves state-of-the-art results.

## 5 CONCLUSION

In this paper, we propose PPBoost, a progressive prompt boosting paradigm that bridges the gap between text-driven VLM and visual-prompted segmentation models for medical image segmentation. PPBoost progressively amplifies weak text-prompted signals into robust visual prompts by (i) training a teacher–student detector on VLM-derived pseudo-bboxes that are filtered by an uncertainty criterion and (ii) introducing a selective bbox expansion strategy to refine visual prompts at inference. PPBoost consistently outperforms other text- and visual-driven segmentation baselines across three diverse medical datasets, and even surpasses few-shot segmentation models without requiring any spatial annotations. Our experiments confirm that PPBoost is a generalizable framework, showing consistent effectiveness across different SSOD backbones and segmentation architectures.

ETHICS STATEMENT

This study develops a text-to-visual prompted medical image segmentation method using only publicly available, de-identified medical image datasets and publicly released foundation models, in accordance with the ICLR Code of Ethics. We did not collect new data, involve human subjects, or access protected health information.

REPRODUCIBILITY STATEMENT

We ensure reproducibility by documenting implementation of training and inference procedures in Experiment section, including backbone models, hyperparameters, optimization, evaluation metrics, and hardware specifications. Dataset descriptions and splitting are detailed in Appendix . An anonymous GitHub repository link provides code, configs, and scripts to reproduce segmentation results. We fix random seeds and report mean performance.

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

# A APPENDIX

## A.1 DATASET DETAILS

**BraTS21** is a large-scale MR glioma segmentation dataset, comprising 1251 3D volumetric cases across four different imaging modalities: T1, T1Gd, T2, and T2-FLAIR. Our experiments focus on the T2 modality, utilizing 6101 training, 760 validation, and 786 test images sliced from different volume cases. **LiTS17** is a CT benchmark for liver and liver tumor segmentation. In this experiment, we evaluate our pipeline in the liver segmentation task, using 7,529 training, 1,140 validation, and 913 test images extracted from 130 volume CT scans. **KidneySeg** is a synthesized ultrasound dataset for the kidney segmentation task, including 3668 training images, and partitioned into 459 validation and 459 test images in our experiment.

## A.2 EMPIRICAL ANALYSIS OF BOUNDING-BOX EXPANSION EFFECT

In this section, we provide empirical evidence to support our refinement strategy of expanding pseudo-bounding boxes. As discussed in Section 2.3, detector-predicted bboxes are often undersized due to noisy supervision from text-driven models. To examine whether uniformly expanding pseudo-bboxes can improve the overall segmentation performance, we conducted a controlled experiment using the widely adopted visual-prompted foundation model MedSAM. To ensure fairness and demonstrate the generalization ability of this analysis beyond our main pipeline, we select six publicly available datasets that cover diverse imaging modalities and anatomical targets: Brain Tumor (MRI) Cheng (2017), CHAOS (CT) Kavur et al. (2021), BUSI (Ultrasound) Al-Dhabyani et al. (2020), Chest X-ray (X-ray) Ninja (2025), ISIC (Dermoscopic) Codella et al. (2019), and EndoCV2021 (Endoscopic) Ali et al. (2021). These datasets are distinct from those used in our PP-Boost experiments, ensuring that the findings reflect the robustness of bbox expansion as a general property of visual-prompted segmentation models rather than a dataset-specific effect.

From each dataset, we sampled 300 image–mask pairs and extracted ground-truth (GT) bounding boxes. We then simulated two types of imperfect pseudo-bboxes through systematic perturbation: undersized boxes, created by shrinking the GT box side lengths by 10%, 15%, and 20%, and oversized boxes, created by expanding them by the same ratios. Both perturbed and original GT boxes were used as prompts for MedSAM, and segmentation performance was evaluated using the mDSC.

The results, summarized in Fig. 4, show that expanding undersized boxes substantially improves segmentation accuracy, recovering much of the lost structure that was omitted in the shrunken prompts. Conversely, expanding already high-quality (or GT) boxes slightly produces little to no degradation, as the added margins contribute minimal noise to MedSAM's segmentation. Fig. 5 provides visual case studies across datasets, illustrating how expansion corrects discriminative region errors in undersized boxes while leaving high-quality predictions largely intact.

These findings confirm that expansion is a robust refinement strategy: even without knowing whether a predicted box is undersized or high-quality, uniformly enlarging pseudo-bboxes with a small ratio yields a net performance gain by compensating for the prevalent undersized cases.

## A.3 USAGE OF LLMS

In this work, large language models (LLMs) are primarily employed as auxiliary tools to enhance the writing. Specifically, we leverage LLMs for two main purposes: (i) refining and polishing the textual presentation to ensure clarity and readability; and (ii) assisting in the development of data visualization code, thereby streamlining the process of transforming experimental results into interpretable figures.

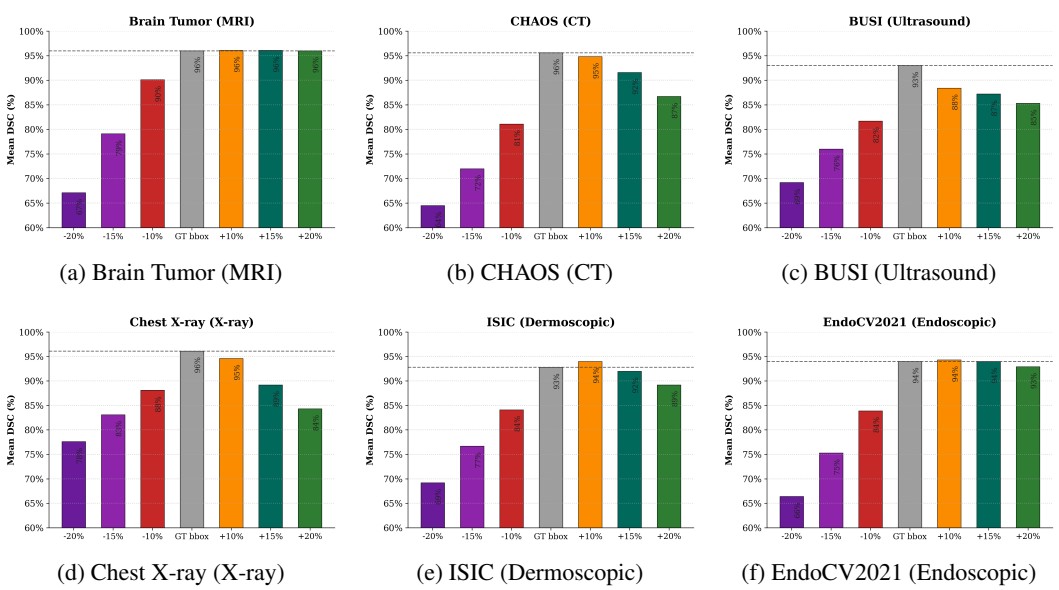

Figure 4: Sensitivity of MedSAM segmentation to bounding-box perturbations. Bars show GT bbox at center with negative (left) and positive (right) perturbations. All plots share a 0–100% y-axis; dotted line marks the GT bbox prompted baseline.

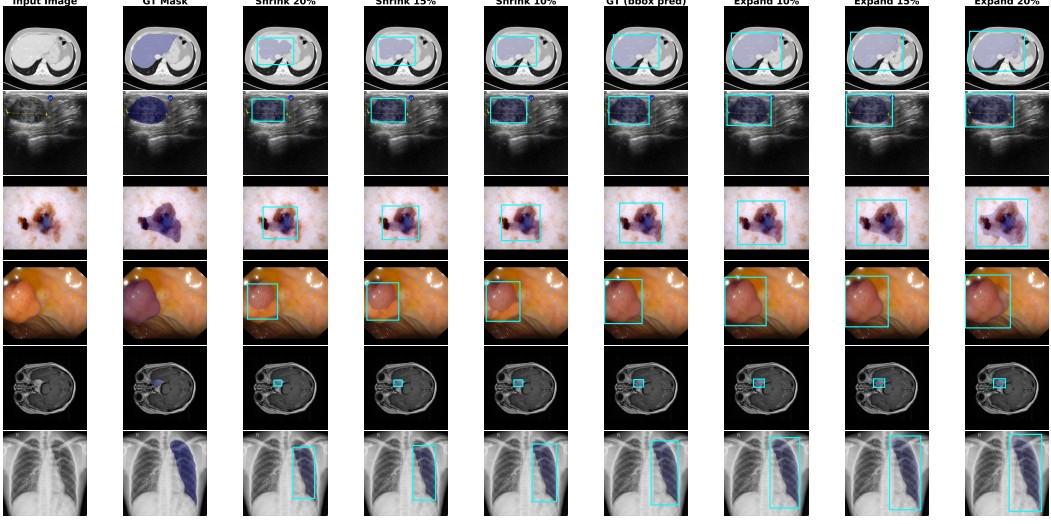

Figure 5: Visualization of the segmentation results on different perturbation levels of bboxes across six datasets