# OpenReview forum: "PPBOOST: PROGRESSIVE PROMPT BOOSTING FOR TEXT-DRIVEN MEDICAL IMAGE SEGMENTATION"
_ICLR.cc/2026/Conference — Submitted to ICLR 2026_

### Official Review · Reviewer_vrGY · 2025-10-27

**Soundness:** 3
**Presentation:** 3
**Contribution:** 2
**Rating:** 4
**Confidence:** 3

**Summary:**

This paper propose PPBoost , a method proposed to tackle zero-shot medical image segmentation by bridging the gap between text prompts and visual prompts. PPBoost progressively transform a natural language description of the target anatomy into a high-quality spatial bounding box, which then guides a segmentation model. PPBoost achieves significantly better segmentation accuracy (measured by mean Dice Similarity and Normalized Surface Distance) than both text-prompted and visual-prompted baselines.

**Strengths:**

1. The paper introduces a creative two-stage pipeline that progressively turns weak text cues into precise visual prompts. This idea of combining a text-based VLM and a detection model to generate segmentation prompts is novel and effectively addresses the core problem of poor spatial grounding in text-only segmentation.
2. The paper provides extensive experimental evidence across three different datasets (different organs and modalities) and shows consistent improvements over baselines.
3. The paper use temperature softmax maps and a KL-divergence criterion to filter out unreliable pseudo-bounding boxes. This *self-ensemble filtering* strategy ensures that only high-confidence pseudo-labels are used to train the detector.

**Weaknesses:**

1. PPBoost’s success hinges on the initial vision-language model (BiomedParse) being able to highlight the target region at least coarsely. If the VLM fails to localize the object (e.g. very small lesions or unseen categories), those cases might be filtered out or produce poor pseudo-labels.
2. The current formulation seems to assume each image has one primary target structure described by one text prompt. It’s unclear how PPBoost handles cases with **multiple instances** of a target or multiple different target classes in the same image. For example, if an image contains two tumors, does the detector output two boxes and can the segmentation model handle multiple prompts?
3. The approach uses a fixed pool of 20 text prompts per dataset, with one randomly assigned to each image. Have the authors evaluated how the *choice of text prompt* affects the outcome?
4. The details of the detector are not clear. The performance is determine the final results of the segmentations. However, the discussion of the detector lack of clarity.

**Questions:**

1. There seems to be some confusion about how the text prompt functions during the inference stage, as the detector (Fast R-CNN) appears to take the original image as input and directly outputs the bounding boxes.
2. There is a potential concern regarding the fairness of the experimental comparisons. Were all the baseline models trained on the same dataset or retrained on the new datasets?
3. Could you clarify why nnU-Net is considered a text-driven segmentation model?

**Details Of Ethics Concerns:**

No Ethics Concerns.

---

### Official Review · Reviewer_YKMW · 2025-10-31

**Soundness:** 3
**Presentation:** 3
**Contribution:** 2
**Rating:** 2
**Confidence:** 4

**Summary:**

The paper operates under a strict zero-shot setting, using a two-stage “text → pseudo-box → segmentation” pipeline to transform weak textual cues into strong spatial prompts. During training, a Vision-Language Model (VLM) based on *BiomedParse* generates confidence maps from textual descriptions, followed by temperature perturbation and KL-divergence filtering to remove unstable samples. The remaining image–box pairs are used to train a semi-supervised detector (Teacher–Student/EMA).  At inference, the detector produces boxes which are then selectively expanded based on confidence, and the refined boxes are used as visual prompts for segmentation models such as MedSAM, SAM, or SAM-Med2D, yielding the final masks.  Across BraTS21, LiTS17, and KidneySeg, PPBoost consistently improves Dice and NSD over text- and visual-prompt baselines, and even surpasses several few-shot segmentation models without using labeled data. The code repository is publicly released.

**Strengths:**

1. From text-based confidence maps to uncertainty filtering and semi-supervised detection, then to selective expansion at inference, PPBoost establishes a reproducible “weak → strong” prompt conversion pipeline that reduces reliance on manual bounding boxes or points.
2. On three heterogeneous datasets, PPBoost achieves mean **mDSC +6.69%** and **mNSD +7.32%** improvements over text/visual-prompted baselines.
3. The authors have made the implementation code publicly available.

**Weaknesses:**

1. **Limited dataset coverage.** The main experiments are conducted only on three datasets (brain tumor MRI, liver CT, and synthetic kidney ultrasound). This limits robustness across anatomy types, institutions, and imaging protocols. The KidneySeg dataset is synthetic, so real-world generalization remains to be validated.
2. **Lack of comparison with related recent works.**
3. **Missing systematic robustness analysis.** Although the authors claim stability across modalities and anatomies, there is no systematic evaluation on **out-of-distribution (OOD)**, noise, or occlusion scenarios, nor categorized failure analysis.

**Questions:**

1. Could you include additional anatomical regions or modalities (e.g., prostate, lung, breast, endoscopy) or **real ultrasound** datasets to validate the generalization and clinical transferability beyond the current three datasets?
2. Could you report **end-to-end training and inference time**, GPU count/hours, and **memory footprint**, as well as achievable **batch size and latency on a 24 GB GPU**, to assess near–real-time feasibility in clinical settings?

---

### Official Review · Reviewer_14ku · 2025-10-31

**Soundness:** 2
**Presentation:** 1
**Contribution:** 1
**Rating:** 4
**Confidence:** 4

**Summary:**

This paper introduces PPBoost, a bridge between VLM and visual prompt-based segmentors to address the challenges that obtaining good visual prompts is costly, especially when dealing with medical images. The detector, the main modification of this framework, is evaluated on different datasets using different backbones, demonstrating better performance. However, the description of the method is unclear, making it difficult to evaluate its contribution.

**Strengths:**

1.The paper includes extensive ablation experiments that validate the effectiveness of the proposed PPBoost framework.
2.Beyond bypassing the need for costly manual prompts, the method offers a substantial boost in inference speed by replacing the computationally heavy VLM with a highly efficient trained detector.

**Weaknesses:**

1.	Clarity on Baseline Comparisons: The exact configuration of the "Direct" baseline—specifically, whether it also incorporates the proposed uncertainty-aware filtering and bounding-box expansion—should be stated more explicitly to ensure a fair comparison. This clarification is critical, as it determines how much of the performance gain is attributable to the core contribution of the detector training stage. A more informative ablation would be to compare against a baseline that uses Filtering and Expansion but no Detector, which would cleanly isolate the detector's contribution.
2.	The proposed PPBoost framework is an effective composition of existing technical blocks (a VLM for initial pseudo-labels, a semi-supervised detector, and a visual-prompted segmenter). While the overall pipeline is well-engineered and validated, the paper would benefit from a more focused discussion on the specific novelty of this composition, as opposed to the incremental improvement of each individual component. A clearer articulation of the unique conceptual leap provided by the framework is encouraged.
3.	While the method operates in a "zero-shot" manner regarding manual spatial annotations, it importantly relies on a substantial number of unlabeled examples to train the detector. This requirement for a sizable data pool for training presents a different kind of practical barrier compared to few-shot methods that require only a handful of labeled examples. The paper would benefit from discussing this trade-off between annotation cost and data collection cost more explicitly.

**Questions:**

1. For the baseline configuration, can you elaborate more on the configuration?

---

### Official Review · Reviewer_VV2e · 2025-11-01

**Soundness:** 3
**Presentation:** 3
**Contribution:** 3
**Rating:** 6
**Confidence:** 3

**Summary:**

The proposed PPBOOST framework is a multi-stage progressive pseudo-label denoising pipeline. First, a pre-trained vision–language model is used to generate an initial pseudo bounding box for the target object from the text prompt. To improve quality, an uncertainty-based filtering is applied: only high-confidence predictions are kept. Using these reliable pseudo-labeled cases, the authors train a teacher–student object detector (semi-supervised) to better localize the target across all images. At inference, this trained detector produces a bounding box for a new image given the text query. The box is then selectively refined to ensure it fully covers the target. Finally, the refined box serves as a visual prompt to a segmentation model.

**Strengths:**

The method is validated on three diverse datasets (brain tumors, liver tumors, kidney lesions), showing significant performance gains over both recent text-prompted and visual-prompted baselines.

the paper is mostly well-written and organized. The technical details (e.g. filtering threshold, network architectures, training schedule) are provided, and an anonymous code link is included for reproducibility.

**Weaknesses:**

1) The proposed pipeline is quite complex, consisting of multiple stages (VLM-based proposal, pseudo-label filtering, semi-supervised detector training, and segmentation with a foundation model). It relies on several pre-trained models and heavy training procedures, which could make it resource-intensive and tricky to reproduce or deploy in practice. This complexity might place a lot of dependencies on the proper tuning of each stage.
2) While the integration of components is novel for this problem, some individual elements are based on existing techniques. The approach builds on known ideas like pseudo-label filtering to handle noise and the teacher–student paradigm for semi-supervised learning.

**Questions:**

please address the concerns mentioned in weakness session.

---

### Meta-Review · Area_Chair_stW5 · 2026-01-04

**Summary:**

This paper proposes the PPBoost framework, a well‑engineered integration of multiple components, including VLM‑based pseudo‑label generation, uncertainty‑aware filtering and expansion, semi‑supervised detector training, and foundation‑model segmentation. Several reviewers (VV2e, 14ku) point to the high complexity and heavy computational dependencies of the pipeline, relying on multiple pre‑trained models and extensive multi‑stage training, which may hinder reproducibility and deployment. The framework builds largely on existing ideas (e.g., pseudo‑label filtering, teacher–student semi‑supervised learning), and its novelty lies mostly in the composition rather than in substantial algorithmic innovations; this conceptual leap is not clearly articulated. Baseline comparisons lack clarity, and the ablation design does not cleanly isolate the detector’s contribution. Practical constraints were also raised (14ku): although the method avoids manual spatial annotations, it needs a large volume of unlabeled data for detector training, a trade‑off with few‑shot methods not fully discussed. Reviewer YKMW highlights limited dataset coverage (three datasets, with one synthetic), absence of recent baseline comparisons, and lack of systematic OOD, noise, or occlusion robustness analysis. Reviewer vrGY additionally notes that PPBoost’s success depends critically on the initial VLM’s localization ability; weaknesses in handling small objects, multiple instances, or multiple target classes in one image are unexplored. The influence of fixed text prompt selection is unquantified, and detector details remain unclear.

Considering these major concerns and the fact that the authors did not provide a response, the AC has decided to reject this paper.

**Reviewer Concerns:**

The authors did not provide any response, and all the concerns are still outstanding.

**Reviewer Scores:**

The authors did not provide any response, and the reviewers have no clear motivation to change scores.

---

### Decision · Program_Chairs · 2026-01-26

Reject